# Cumulative incidence, prevalence, seroconversion, and associated factors for SARS-CoV-2 infection among healthcare workers of a University Hospital in Bogotá, Colombia

**Sandra Liliana Valderrama-Beltrán**[1,2]\*, **Juliana Cuervo-Rojas**[3], **Beatriz Ariza**[4], **Claudia Cardozo**[4], **Juana Ángel**[5], **Samuel Martinez-Vernaza**[2], **María Juliana Soto**[2], **Julieth Arcila**[4], **Diana Salgado**[4], **Martín Rondón**[3], **Magda Cepeda**[3], **Julio Cesar Castellanos**[6], **Carlos Gómez-Restrepo**[6], **Manuel Antonio Franco**[5]\*

1 PhD Program in Clinical Epidemiology, Department of Clinical Epidemiology and Biostatistics, Faculty of Medicine, Pontificia Universidad Javeriana, Bogotá, Colombia, 2 Division of Infectious Diseases, Department of Internal Medicine, Faculty of Medicine, Pontificia Universidad Javeriana, Hospital Universitario San Ignacio Infectious Diseases Research Group, Bogotá, Colombia, 3 Department of Clinical Epidemiology and Biostatistics, Faculty of Medicine, Pontificia Universidad Javeriana, Bogotá, Colombia, 4 Clinical Laboratory Science Research Group, Clinical Laboratory, Hospital Universitario San Ignacio, Bogotá, Colombia, 5 Institute of Human Genetics, Faculty of Medicine, Pontificia Universidad Javeriana, Bogotá, Colombia, 6 Faculty of Medicine, Pontificia Universidad Javeriana, Hospital Universitario San Ignacio, Bogotá, Colombia

\* slvalderrama.med@javeriana.edu.co (SLVB); mafranco@javeriana.edu.co (MAF)

**Data Availability Statement:** All relevant data are within the paper and its Supporting Information files.

## Abstract

This study aimed to determine the cumulative incidence, prevalence, and seroconversion of severe acute respiratory syndrome coronavirus-2 (SARS-CoV-2) and its associated factors among healthcare workers (HCWs) of a University Hospital in Bogotá, Colombia. An ambispective cohort was established from March 2020 to February 2021. From November 2020 to February 2021, SARS-CoV-2 antibodies were measured on two occasions 14–90 days apart to determine seroprevalence and seroconversion. We used multivariate log-binomial regression to evaluate factors associated with SARS-CoV-2 infection. Among 2,597 HCWs, the cumulative incidence of infection was 35.7%, and seroprevalence was 21.5%. A reduced risk of infection was observed among those aged 35–44 and ≥45 years (adjusted relative risks [aRRs], 0.84 and 0.83, respectively), physicians (aRR, 0.77), those wearing N95 respirators (aRR, 0.82) and working remotely (aRR, 0.74). Being overweight (aRR, 1.18) or obese (aRR, 1.24); being a nurse or nurse assistant (aRR, 1.20); working in the emergency room (aRR, 1.45), general wards (aRR, 1.45), intensive care unit (aRR, 1.34), or COVID-19 areas (aRR, 1.17); and close contact with COVID-19 cases (aRR, 1.47) increased the risk of infection. The incidence of SARS-CoV-2 infection found in this study reflects the dynamics of the first year of the pandemic in Bogotá. A high burden of infection calls for strengthening prevention and screening measures for HCWs, focusing especially on those at high risk.

**Funding:** This study was funded by Pontificia Universidad Javeriana, Hospital Universitario San Ignacio, and Fundación Bolívar Davivienda. The funders had no role in study design, data collection and analysis, decision to publish, or preparation of the manuscript.

**Competing interests:** The authors have declared that no competing interests exist.

## Introduction

In previous epidemics of coronaviruses and the severe acute respiratory syndrome coronavirus 2 (SARS-CoV-2) pandemic, healthcare workers (HCWs) have been recognized as a population with a high risk of infection, transmission, and propagation of the disease [1, 2]. These risks are related to increased occupational exposure in addition to the exposure in their communities [3, 4]. The reported prevalence of SARS-CoV-2 infection and coronavirus disease (COVID-19) in HCWs range from 3% to 51% [1, 5–7]. This wide range reflects not only the different epidemic periods in which the assessments were conducted but also the difference in the exposure risks in the communities, availability of personal protective equipment (PPE), and other region-associated social and economic conditions that define the risk of exposure [3, 6, 8].

In HCWs, several risk factors for having antibodies against SARS-CoV-2 have been consistently described, such as working in COVID-19 wards, having close contact with COVID-19 cases, inadequate use of PPE, or having social reunions outside of the workplace [9, 10]. However, differences in seroprevalence by age, sex, occupation, or body mass index (BMI) remain unclear, considering that some studies showed contradictory results or had incomplete data [6, 11].

The COVID-19 pandemic hit Latin America (LATAM) in late February 2020. As of February 2022, 6,026,988 cases were reported in Colombia (117,429 cases per million inhabitants), with a third of the cases reported in Bogotá, the country's capital [12]. The cumulative mortality rate in Colombia was 2686 deaths per million inhabitants by March 2022, which was the 23rd highest mortality rate in the world [13].

Even though several studies have assessed the burden of SARS-CoV-2 infection in HCWs, data from low- and middle-income countries such as Colombia are scarce, especially concerning the incidence of SARS-CoV-2 and associated factors. In a previous study conducted in 10 major cities in Colombia, the seroprevalence of SARS-CoV-2 in the HCWs was 35% from September to November 2020 [14].

In this study, we determined the cumulative incidence of SARS-CoV-2 infection from March 2020 to February 2021, the seroprevalence from November 2020 to February 2021, the prevalence of acute infection, seroconversion, and factors associated with infection among the HCWs of Hospital Universitario San Ignacio (HUSI), a tertiary referral care institution in Bogotá, Colombia.

## Methods

### Study design

An ambispective cohort of HCWs from HUSI in Bogotá—Colombia was followed from March 2020 to February 2021. The cohort had two components: a prospective one with a baseline seroprevalence study, and a retrospective component.

All HCWs employed by HUSI, including administrative staff, were invited to participate by an institutional e-mail. In the first component, to determine SARS-CoV-2 seroprevalence, blood samples were drawn between November 17, 2020 and February 12, 2021. In addition, a subsample of 703 HCWs with no documented history of SARS-CoV-2 infection was randomly selected to determine the point prevalence of acute infection using the results of reverse transcription polymerase chain reaction (RT-PCR) tests of nasopharyngeal swab samples collected between November 17 and November 30, 2020.

The participants were asked to complete a web-based questionnaire about sociodemographic, clinical, and occupational characteristics and history of SARS-CoV-2 infection

confirmed by RT-PCR results, antigen tests for SARS-CoV-2 spike protein, or IgG antibodies and associated symptoms. The questionnaire was designed in REDCap™ (Research Electronic Data capture) [15].

To evaluate seroconversion and seropermanence, the participants were followed prospectively, with a second blood sample collected 14–90 days after the first one, in the period from December 15, 2020 to February 26, 2021.

For the second component of the study, we assembled a retrospective cohort from March 6, 2020 (first confirmed COVID-19 case in the city) to November 16, 2020 involving those who agreed to participate in the first component (prevalence study). For this purpose, the information collected in the web-based questionnaire was linked with data from a registry of all documented SARS-CoV-2 infections among the HCWs that is maintained by HUSI's occupational health office; the registry includes sociodemographic, clinical, and occupational data of the cases, and the method of diagnosis (RT-PCR and antigen tests mainly for symptomatic patients who sought care or IgG antibody tests for screening personnel working in COVID-19 wards). We also linked results from IgG tests performed in the HCWs in a previous study [16].

This study followed the Strengthening the Reporting of Observational Studies in Epidemiology (STROBE) reporting guideline [17]. The study and informed consent form were approved by the Ethics Committee of the School of Medicine of the Pontificia Universidad Javeriana (PUJ) and HUSI (FM-CIE-0686-21). Inform consent was obtained from all participants.

## Laboratory methods

The primary serological test used was a hemagglutination assay (HA), as described below. For a subsample of HCWs with a positive HA result additional commercial tests were performed: IgM determination by Enzyme Linked Fluorescence Assay (ELFA) and IgG determination by a chemiluminescent assay (CLIA). If the results of these two tests were negative and there was no documented history of SARS-CoV-2 infection, IgG antibodies specific for the receptor binding domain (RBD) of the S protein were measured with an enzyme-linked immunosorbent assay (ELISA) (S1 Methods).

**RBD HA.** This test detects antibodies against the RBD of the SARS-CoV-2 spike protein (S1 Methods; S1 Fig). The reported sensitivity is 90% and specificity is 99% for detection of antibodies after an RT-PCR-diagnosed infection [18]. The assay was performed as described by Townsend et al [18].

**RT-PCR.** RT-PCR was performed in the accredited HUSI Clinical Laboratory using nasopharyngeal swabs samples or aspirates collected using the VIASURE™ Real-Time PCR Detection Kit plates (CerTest BIOTEC, Zaragoza, Spain).

## Definitions

Seropositivity in the first component (seroprevalence study and prospective cohort), was defined as a confirmed positive HA result. Confirmation was based on any of the following: (i) another positive result for any of the following tests: IgM ELFA, IgG CLIA, or IgG ELISA; (ii) two consecutives positive HA results; or (iii) a history of SARS-CoV-2 infection. HCWs who had only one positive HA assessment with no availability of serum to perform ELISA after negative results on IgM ELFA and IgG CLIA were also classified as seropositive (S2 Fig).

For the prospective cohort, the definition of seroconversion was the appearance of a positive HA, and the definition of seropermanence was the persistence of a positive HA at the end of the follow-up period.

For evaluating the point prevalence of acute infection, a positive RT-PCR was confirmed if the HCW had symptoms. An asymptomatic HCW was considered acutely infected if he/she had a

positive RT-PCR result and any of the following conditions: negative HA, positive HA and positive IgM on ELFA, or positive HA with negative IgM on ELFA and negative IgG by CLIA (S3 Fig).

We defined a case of infection by SARS-CoV-2 as a HCW who had a confirmed infection either by RT-PCR, IgG antibody, or antigen testing in the retrospective study, who was classified as seropositive in the seroprevalence study, or who had a positive RT-PCR result in the acute infection prevalence study.

## Statistical analysis

The data were analyzed in R software version 4.1 (R Project for Statistical Computing). We used the *logbin* package [19]. Initially, extreme values, possible digitation errors, and missing data were evaluated. In such cases, data were confirmed either by HUSI's institutional registries or completed by a follow up call.

We conducted a descriptive analysis of the demographic characteristics of the study participants. Continuous variables were described using medians and interquartile ranges (IQRs), and categorical variables were described using absolute and relative frequencies.

To calculate the SARS-CoV-2 cumulative incidence, the number of SARS-CoV-2 cases was divided by the number of study participants. To calculate the cumulative seroconversion incidence for the prospective cohort, we divided the number of seropositive cases at follow-up by the number of seronegative cases at baseline, based on the data of those whose first and second blood samples were available.

We evaluated the association between being a case of SARS-CoV-2 and the sociodemographic, clinical, and occupational characteristics. We estimated the relative risks (RR) using multivariate log-binomial regression models. We did not find any collinear variables in the model diagnosis; all variance inflation factor (VIF) values were inferior to 3. As missing data affected only about 6% of the records, we decided to conduct a complete-subject analysis.

We evaluated the sensitivity of the results by conducting an additional analysis in which we defined any individual with positive HA as a seropositive case.

## Results

The study enrolled 2,597 HCWs, 79.1% of the 3,282 that constituted the target population. The median age of the study population was 34.2 years (IQR, 28.3–41.6), and females accounted for 74.7% of the population (n = 1,940). Most HCWs provided direct patient care (n = 2,026) and the majority were nurse assistants (n = 674), nurses (n = 326) or specialist physicians (n = 327). The complete characteristics of the population are described in Table 1. The distributions by sex, age, or type of occupation were similar among workers who participated and those who did not participate (S1 Table).

### Acute infection

Among the 703 HCWs who underwent a SARS-CoV-2 RT-PCR test between November 17 and November 30, 2020, 20 fulfilled the definition of acute infection, resulting in a point prevalence of 2.8% (99% confidence interval, 1.2%–4.5%). Further, 55% of the HCWs with acute infections were asymptomatic (n = 11).

### Seroprevalence

Between November 17, 2020 and February 12, 2021, baseline antibodies were detected in 23.4% of the HCWs (n = 607/ 2,597) by HA. However, only 558 were confirmed seropositive cases according to the algorithm for definition of seropositive cases (S2 Fig), resulting in a

**Table 1. Characteristics of the study population in Hospital Universitario San Ignacio (Retrospective cohort and seroprevalence study).** March 6, 2020 to February 12, 2021.

| Participant Characteristics | n (%)[1] |
|---|---|
| **Sex (n = 2597)** | |
| Female | 1940 (74.7%) |
| Male | 657 (25.3%) |
| **Age (years), (n = 2597)** | |
| < 35 | 1377 (53.0%) |
| 35–44 | 772 (29.7%) |
| ≥ 45 | 448 (17.3%) |
| **Type of occupation (n = 2597)** | |
| Direct patient care | 2026 (78.0%) |
| Administrative | 571 (22.0%) |
| **Type of direct patient care worker (n = 2026)** | |
| Medical specialist | 327 (16.1%) |
| Resident | 242 (11.9%) |
| General physician | 49 (2.4%) |
| Nurse | 326 (16.1%) |
| Nurse assistant | 674 (33.3%) |
| Bacteriologist | 67 (3.3%) |
| Respiratory therapist | 30 (1.5%) |
| Nutritionist | 10 (0.5%) |
| Other | 301 (14.9%) |
| **Main Service (n = 2585)** | |
| Administrative departments | 373 (14.4%) |
| Emergency room | 392 (15.2%) |
| General wards | 711 (27.5%) |
| ICU[2] | 266 (10.3%) |
| Surgical Areas | 285 (11.0%) |
| Ambulatory and diagnostic services | 558 (21.6%) |
| **Adequate use of PPE[3] (n = 2548)** | |
| Yes | 2482 (97.4%) |
| No | 66 (2.6%) |
| **Type of respiratory protection (n = 2536)** | |
| Cloth mask | 100 (3.9%) |
| Surgical mask | 1006 (39.7%) |
| N-95 respirator | 1430 (56.4%) |
| **History of close contact[4] (n = 2523)** | |
| Yes | 1219 (48.3%) |
| No | 1304 (51.7%) |
| **Type of close contact[5] (n = 1215)** | |
| Outside of the work environment | 289 (23.8%) |
| Work area | 863 (71.0%) |
| HUSI wellness area | 63 (5.2%) |
| **Type of work (n = 2548)** | |
| Remote work | 304 (11.9%) |
| Non-remote work | 2244 (88.1%) |
| **COVID-19 work[6] (n = 2546)** | |
| Yes | 1342 (52.7%) |

*(Continued)*

**Table 1.** (Continued)

| | |
|---|---|
| **No** | 1204 (47.3%) |
| **Shift (n = 2549)** | |
| Day shift | 1690 (66.3%) |
| Night shift | 859 (33.7%) |
| **Type of transportation[7] (n = 2526)** | |
| Unshared | 1235 (48.9%) |
| Shared | 1291 (51.1%) |
| **Work in more than one institution (n = 2540)** | |
| Work at only one institution | 2301 (90.6%) |
| Work at two or more institutions | 239 (9.4%) |
| **Smoking in the previous year[8] (n = 2516)** | |
| Yes | 317 (12.6%) |
| No | 2199 (87.4%) |
| **Influenza vaccination in the previous year[9] (n = 2482)** | |
| Yes | 974 (39.2%) |
| No | 1508 (60.8%) |
| **Body Mass Index (kg/m2), (n = 2514)** | |
| Low or normal ($<25$) | 1503 (59.8%) |
| Overweight (25.0–29.9) | 826 (32.9%) |
| Obesity ($\geq 30$) | 185 (7.4%) |
| **Comorbidities[10,11] (n = 2525)** | |
| Any comorbidity | 427 (16.9%) |
| Arterial Hypertension | 129 (5.1%) |
| Hypothyroidism | 111 (4.4%) |
| Asthma | 88 (3.5%) |
| Autoimmune Disease | 30 (1.2%) |
| Cancer | 12 (0.5%) |

[1]Column-based percentages.

[2]ICU, Intensive Care Unit.

[3]PPE, personal protective equipment. Complete use of PPE since March 2020.

[4]HCWs who were less than 6 feet away from a SARS-CoV-2-infected person (laboratory-confirmed or a clinical diagnosis) for a total of 15 min without PPE, at any time since March 2020.

[5]This category only applies to HCWs with close contact history.

[6]HCWs who worked in the COVID area any time since March 2020.

[7]Shared transportation was defined as the use of any public or collective transport.

[8]History of smoking in the previous year.

[9]History of influenza vaccination in the previous year.

[10]Self-reported pre-existing medical condition.

[11]These categories are not mutually exclusive. HCW, healthcare worker.

seroprevalence of 21.5%. Table 2 shows the seroprevalence according to the characteristics of the HCWs.

A second blood sample was obtained from 1,654 (63.7%) out of the initial 2,597 HCWs between December 15, 2020 and February 26, 2021, at a median follow-up time of 39 days (IQR, 32–49). In this group of HCWs, 27.4% (n = 453) had positive antibodies on HA but only 24.8% (n = 410) were confirmed seropositive cases according to the algorithm (S2 Fig). In Fig 1. we illustrate the SARS-CoV-2 seroprevalence and cumulative incidence among HCWs of HUSI, and Bogotá's COVID-19 epidemic curve from March 2020 to February 2021.

**Table 2. Seroprevalence of SARS-CoV-2 by characteristics of the HCWs.** Hospital Universitario San Ignacio, November 17, 2020–February 12, 2021.

| Participant Characteristics | Total (n) | Prevalence (n (%)[1]) |
|---|---|---|
| Sex | | |
| Female | 1940 | 426 (22.0%) |
| Male | 657 | 132 (20.1%) |
| Age category (years) | | |
| < 35 | 1377 | 320 (23.2%) |
| 35–44 | 772 | 161 (20.9%) |
| ≥ 45 | 448 | 77 (17.2%) |
| Type of occupation | | |
| Administrative | 571 | 108 (18.9%) |
| Physician | 618 | 90 (14.6%) |
| Nurse | 1000 | 288 (28.8%) |
| Other | 408 | 72 (17.6%) |
| Main Service | | |
| Administrative office | 373 | 60 (16.1%) |
| Emergency room | 392 | 104 (26.5%) |
| General wards | 711 | 191 (26.9%) |
| ICU[2] | 266 | 62 (23.3%) |
| Surgical Areas | 285 | 43 (15.1%) |
| Ambulatory and diagnostic services | 558 | 97 (17.4%) |
| Type of work | | |
| Remote work | 304 | 50 (16.4%) |
| Non-remote work | 2244 | 504 (22.5%) |
| COVID-19 work[3] | | |
| Yes | 1342 | 328 (24.4%) |
| No | 1204 | 225 (18.7%) |
| Shift | | |
| Day shift | 1690 | 347 (20.5%) |
| Night shift | 859 | 206 (24.0%) |
| Type of respiratory protection | | |
| Cloth mask | 100 | 17 (17.0%) |
| Surgical mask | 1006 | 216 (21.5%) |
| N-95 respirator | 1430 | 317 (22.2%) |
| History of close contact[4] | | |
| Yes | 1219 | 311 (25.5%) |
| No | 1304 | 239 (18.3%) |
| Type of transportation[5] | | |
| Unshared | 1235 | 240 (19.4%) |
| Shared | 1291 | 308 (23.9%) |
| Work in more than one institution | | |
| Work at only one institution | 2301 | 515 (22.4%) |
| Work at two or more institutions | 239 | 35 (14.6%) |
| Smoking in the previous year[6] | | |
| Yes | 317 | 71 (22.4%) |
| No | 2199 | 476 (21.6%) |
| Influenza vaccination in the previous year [7] | | |
| Yes | 974 | 201 (20.6%) |

*(Continued)*

**Table 2.** (Continued)

| Participant Characteristics | Total (n) | Prevalence (n (%)[1]) |
|---|---|---|
| No | 1508 | 331 (21.9%) |
| Body Mass Index (kg/m$^2$) | | |
| Low or normal (<25) | 1503 | 300 (20.0%) |
| Overweight (25.0–29.9) | 826 | 190 (23.0%) |
| Obesity (≥30) | 185 | 57 (30.8%) |
| Comorbidities[8] | | |
| Any comorbidity | 427 | 86 (20.1%) |
| Non-comorbidity | 2098 | 466 (22.2%) |
| History of Arterial Hypertension | | |
| Yes | 129 | 27 (20.9%) |
| No | 2396 | 525 (21.9%) |
| History of Hypothyroidism | | |
| Yes | 111 | 26 (23.4%) |
| No | 2414 | 526 (21.8%) |
| History of Asthma | | |
| Yes | 88 | 15 (17.0%) |
| No | 2437 | 537 (22.0%) |
| History of Autoimmune Disease | | |
| Yes | 30 | 4 (13.3%) |
| No | 2495 | 548 (22.0%) |
| History of Cancer | | |
| Yes | 12 | 2 (16.7%) |
| No | 2513 | 550 (21.9%) |

[1]Row-based percentages.

[2]ICU, Intensive Care Unit.

[3]HCW who has worked in the COVID area any time since March 2020.

[4]HCW who was less than 6 feet away from a SARS-CoV-2-infected person (laboratory-confirmed or a clinical diagnosis) for a total of 15 min without personal protective equipment, at any time since March 2020.

[5]Shared transportation was defined as the use of any public or collective transport.

[6]History of smoking in the previous year.

[7]History of influenza vaccination in the previous year.

[8]Self-reported pre-existing medical condition. HCW, healthcare worker.

## Seroconversion and seropermanence

The median age of the 1,654 HCWs who were included in the second antibody assessment was 36.3 years (IQR 30.3–43.4). Those who did not return for the follow-up were more frequently younger than 35 years (68.0% *vs*. 40.9%), male (32.9% *vs*. 21.0%), and worked directly with patients (82.4% *vs*. 75.3%) than those who returned for the second blood exam (S2 Table).

Among those with a follow-up sample who were seronegative at baseline, 12.3% (164/1,338) seroconverted. Regarding seropermanence, 77.8% (246/316) of the HCWs who were seropositive at baseline remained seropositive at follow-up.

## History of symptomatic and asymptomatic SARS-CoV-2 infection

From March 6, 2020 to February 12, 2021 (retrospective cohort up to the time of the seroprevalence study), 28.9% (750/2,597) HCWs had a history of prior infection diagnosed by RT-PCR

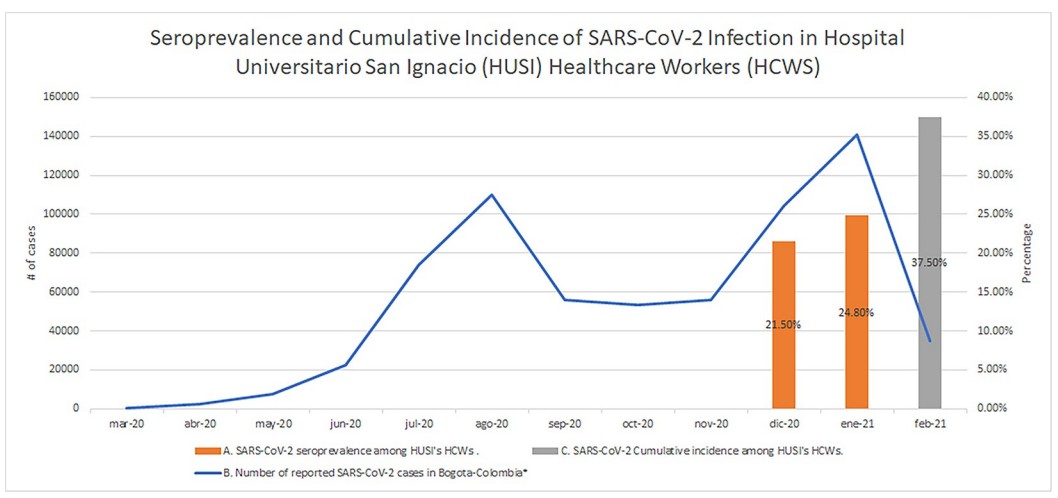

**Fig 1. Seroprevalence and cumulative incidence of SARS-CoV-2 infection in Hospital Universitario San Ignacio (HUSI) healthcare workers (HCWs), and Bogotá's COVID-19 epidemic curve.** A. The SARS-CoV-2 seroprevalence in HUSI HCWs was 21.5% between November 17, 2020 and February 12, 2021 (n = 2,597) and 24.8% (n = 1,654) between December 15, 2020 and February 26, 2021. B. The SARS-CoV-2 epidemic curve of Bogotá between March 2020 and February 2021 shows two epidemic waves. The first one began in June 2020 and ended approximately in October 2020, and the second one began in November 2020 and ended in February 2021. In this last epidemic wave, Gamma (P.1) and Mu (B.1.621) variants were introduced in the city. C. The SARS-CoV-2 cumulative incidence in HUSI HCWs was 35.7% (927/2,597) between March 6, 2020 and February 12, 2021. *The numbers of SARS-CoV-2 infection cases were taken from: https://saludata.saludcapital.gov.co/osb/index.php/datos-de-salud/enfermedades-trasmisibles/covid19/.

in 83.6% (n = 627), by antigen testing in 2.5% (n = 19), and IgG antibody testing in 13.9% (n = 104). Among those with infection confirmed by RT-PCR, 13.6% (85/627) did not report any symptoms.

Among those who did report symptoms (n = 542), the most frequent were headache at 80.4% (n = 436), upper respiratory symptoms at 80.3% (n = 435), fatigue at 77.7% (n = 421), anosmia or dysgeusia at 53.7% (n = 291), myalgia at 49.4% (n = 268), fever at 40.2% (n = 218), lower respiratory symptoms at 29.9% (n = 162), and diarrhea at 23.4% (n = 127).

In the population with a history of infection, the median duration of symptoms was 10 days (IQR, 7–15), 6.5% (49/750) required hospitalization, and no HCWs died.

## Cumulative incidence of infection

The cumulative incidence of infection between March 6, 2020 and February 12, 2021 was 35.7% (927/2,597), according to the case definition presented above. In the multivariate analysis, being overweight (aRR, 1.18) or obese (aRR, 1.24), being a nurse or nurse assistant (aRR, 1.20), working in the emergency room (aRR, 1.45), general wards (aRR, 1.45), intensive care unit (ICU) (aRR, 1.34) or COVID-19 areas (aRR, 1.17); and previous close contact with COVID-19 patients (aRR, 1.47) were associated with an increased risk of SARS-CoV-2 infection. Age 35–44 years (aRR, 0.84) or ≥45 years (aRR, 0.83), in contrast to age <35 years, being a physician (aRR, 0.77), wearing of an N95 respirator (aRR, 0.82) and working remotely (aRR, 0.74) were associated with decreased risk of infection (Table 3). In the sensitivity analysis, where the presence of a positive HA result was defined as a seropositive case, similar results were obtained (Table 3).

## Discussion

Approximately a third of the HCW population (35.7%) was infected in the first year of the pandemic, and a fifth (21.5%) was found to be seropositive between November 17, 2020 and

**Table 3. Association between sociodemographic, clinical, and occupational characteristics and SARS-CoV-2 infection in HCWs at Hospital Universitario San Ignacio from March 6, 2020, to February 12, 2021.**

| Participant Characteristics | SARS-CoV-2 Cumulative Incidence (n, (%))[1] | Crude RR | Adjusted RR[2] | Sensitivity analysis[3] (Adjusted RR) |
|---|---|---|---|---|
| **Sex** | | | | |
| Female (n = 1940) | 693 (35.7%) | ref | ref | ref |
| Male (n = 657) | 234 (35.6%) | 0.99 | 1.05 | 1.07 |
| **Age category (years)** | | | | |
| <35 (n = 1377) | 546 (39.7%) | ref | ref | ref |
| 35–44 (n = 772) | 252 (32.6%) | 0.82 | 0.84 | 0.85 |
| ≥ 45 (n = 448) | 129 (28.8%) | 0.73 | 0.83 | 0.83 |
| **Type of occupation** | | | | |
| Administrative (n = 571) | 153 (26.8%) | ref | ref | ref |
| Physician (n = 618) | 177 (28.6%) | 1.07 | 0.77 | 0.78 |
| Nurse or nurse assistant (n = 1000) | 478 (47.8%) | 1.78 | 1.20 | 1.17 |
| Other (n = 408) | 119 (29.2%) | 1.09 | 0.90 | 0.88 |
| **Main Service** | | | | |
| Administrative office (n = 373) | 86 (23.1%) | ref | ref | ref |
| Emergency room (n = 392) | 173 (44.1%) | 1.91 | 1.45 | 1.41 |
| General wards (n = 711) | 332 (46.7%) | 2.03 | 1.45 | 1.38 |
| ICU[4] (n = 266) | 110 (41.4%) | 1.79 | 1.34 | 1.28 |
| Surgical areas (n = 285) | 76 (26.7%) | 1.16 | 1.04 | 1.04 |
| Ambulatory and diagnostic services (n = 558) | 148 (26.5%) | 1.16 | 1.09 | 1.10 |
| **Type of work** | | | | |
| Non-remote work (n = 2244) | 858 (38.2%) | ref | ref | ref |
| Remote work (n = 304) | 64 (21.1%) | 0.55 | 0.74 | 0.85 |
| **COVID-19 work[5]** | | | | |
| No (n = 1204) | 348 (28.9%) | ref | ref | ref |
| Yes (n = 1342) | 572 (42.6%) | 1.47 | 1.17 | 1.17 |
| **Shift** | | | | |
| Day shift (n = 1690) | 561 (33.2%) | ref | ref | ref |
| Night shift (n = 859) | 361 (42.0%) | 1.27 | 1.07 | 1.07 |
| **Type of respiratory protection** | | | | |
| Surgical or cloth mask (n = 1106) | 386 (34.9%) | ref | ref | ref |
| N-95 respirator (n = 1430) | 529 (37.0%) | 1.06 | 0.82 | 0.83 |
| **History of close contact[6]** | | | | |
| No (n = 1304) | 348 (26.7%) | ref | ref | ref |
| Yes (n = 1219) | 569 (46.7%) | 1.75 | 1.47 | 1.46 |
| **Type of transportation[7]** | | | | |
| Unshared (n = 1235) | 415 (33.6%) | ref | ref | ref |
| Shared (n = 1291) | 497 (38.5%) | 1.15 | 0.97 | 0.98 |
| **Smoking in the previous year[8]** | | | | |
| No (n = 2199) | 788 (35.8%) | ref | ref | ref |
| Yes (n = 317) | 123 (38.8%) | 1.08 | 0.94 | 0.92 |
| **Influenza vaccination in the previous year[9]** | | | | |
| No (n = 1508) | 541 (35.9%) | ref | ref | ref |
| Yes (n = 974) | 342 (35.1%) | 0.98 | 0.94 | 0.96 |
| **Body Mass Index (kg/m$^2$)** | | | | |
| Low or normal (<25), (n = 1503) | 501 (33.3%) | ref | ref | ref |
| Overweight (25.0–29.9), (n = 826) | 324 (39.2%) | 1.18 | 1.18 | 1.15 |

*(Continued)*

**Table 3.** (Continued)

| Participant Characteristics | SARS-CoV-2 Cumulative Incidence (n, (%))[1] | Crude RR | Adjusted RR[2] | Sensitivity analysis[3] (Adjusted RR) |
|---|---|---|---|---|
| Obesity (>30), (n = 185) | 81 (43.8%) | 1.31 | 1.24 | 1.20 |
| Comorbidities[10] | | | | |
| Non-comorbidity (n = 2098) | 771 (36.7%) | ref | ref | ref |
| Any comorbidity (n = 427) | 146 (34.2%) | 0.93 | 0.99 | 0.97 |

[1]Row-based percentages.

[2]aRR: adjusted relative risk. Results from multivariable analysis using log-binomial regression (n = 2,442).

[3]We evaluated the sensitivity of the results to the definition of seropositivity as a confirmed positive HA result with an additional analysis in which we defined as seropositive any positive HA result. This analysis increased the incident cases from 927 (35.7%) to 976 (37.6%). Here, we present the results of the multivariate sensitivity analysis using log-binomial regression (n = 2,442).

[4]ICU, Intensive Care Unit.

[5]HCWs who worked in the COVID area any time since March 2020.

[6]HCWs who were less than 6 feet away from an infected person (laboratory-confirmed or a clinical diagnosis) for a total of 15 min without personal protective equipment, at any time since March 2020.

[7]Shared transportation was defined as the use of any public or collective transport.

[8]History of smoking in the previous year.

[9]History of influenza vaccination in the previous year.

[10]Self-reported pre-existing medical condition. HCW, healthcare worker.

February 12, 2021. The seroprevalence reported globally range between 3% and 51%, increasing with the progression of the pandemic [1, 5–7]. In our study, the seroprevalence was lower than that reported for Bogotá's population (29.9%) [20] and that previously reported for HCWs in Bogotá (34%) [14] in studies with similar time frames as our study. This may indicate that in healthcare institutions with proper availability of PPE and adherence to its use, as in ours (97.4%), HCWs may have an infection risk similar to or lower than that in the community. However, we used a serologic test that may perform differently from the tests used in the studies mentioned above.

When evaluating the point prevalence of acute infection in a random sample of the HCWs, 55% of the cases identified were asymptomatic, similar to what have been reported elsewhere [21, 22]. This result is coherent with the value we have previously reported for asymptomatic infection in HCWs of our hospital [16]. Asymptomatic–either pre-symptomatic or asymptomatic–SARS-CoV-2 infection likely plays an important role in disseminating viral infection [23]. This indicates the importance of considering screening for SARS-CoV-2 for the containment of healthcare-associated outbreaks. However, considering the low prevalence of acute infection (2.8%) and the cost related to this practice, it would be ideal to use this intervention for high-risk HCWs, such as personnel in COVID-19 areas or emergency rooms.

Our cumulative incidence of seroconversion from December 15, 2020 to February 26, 2021 was 12.3%. Such a high incidence might be explained considering the spread of the second SARS-CoV-2 epidemic wave in this time period, where the concurrences of loosening of public health preventive measurements, social exhaustion, and emergence of new SARS-CoV-2 variants in the country, such as Gamma (P.1) and Mu (B.1.621) could have contributed to the higher transmission of the disease [24, 25].

If we consider all the measurements of our cohort (cumulative incidence up to the first round of antibody measurement plus the seroconversion follow-up) around 45% of our HCWs were infected by the end of February 2021, which is slightly lower than the incidence estimated for Bogotá [25]. In Bogotá the SARS-CoV-2 incidence was lower in HCWs than in other populations such as essential service workers, police officials, military forces, or people

with low socioeconomic status [26], which could be related to better standards of living and stricter preventive measurements for HCWs.

We found that HCWs aged ≥35 years had a lower risk of SARS-CoV-2 infection than those aged <35 years, probably due to higher concern regarding possible unfavorable outcomes of infection leading to higher compliance with preventive measures in that age group [27, 28]. A similar trend has been reported in some studies [10, 29, 30], whereas other suggest older ages are more likely to be infected [11, 31].

We explored the occupation of HCWs as a possible risk factor associated with SARS-CoV-2. In our study, being a nurse increased the risk of SARS-CoV-2 infection by 20% compared with being an administrative HCWs, which is consistent with the results of studies in different countries [32–34]. We found a lower risk of infection in physicians than in administrative HCWs, which is unusual from the trend in the literature [35]. Nurses tend to have more prolonged and frequent contact with patients with SARS-CoV-2 infection, and higher rates of burn out than other HCWs, which may lead to more exposure and less compliance with preventive measures, and consequently higher infection rates [36]. Socioeconomic factors that we could not evaluate might contribute to explain these findings.

The personnel working in COVID-19 areas, emergency rooms, ICU, or general wards had a greater risk of SARS-CoV-2 infection than those working in other services of the hospital. This might be explained by the high exposure time to patients with COVID-19 in scenarios with inadequate ventilation. These findings are consistent with the results from numerous studies [1, 9, 37–39].

A noteworthy finding was that being overweight or obese was associated with an increased risk of becoming infected. Obesity has been commonly described as a predictor of severe disease and death, but not as a risk factor for SARS-CoV-2 infection. However, a study in South Africa also found that these conditions in the HCWs were associated with SARS-CoV-2 infection in the multivariate analysis [40]. The reason for this association is yet to be established; however, it could be related to a high susceptibility to respiratory viral infections due to alterations in adaptive or innate immunity [41]. This phenomenon has been described in the immunological response to the influenza virus in obese patients [42].

We observed a reduction in the risk of SARS-CoV-2 infection in the HCWs who used N-95 respirators compared with the risk in those who used surgical or cloth masks, which has been reported previously [32, 43]. In addition, recent research regarding airborne transmission suggests that N95 respirators may be preferable for all HCWs activities [44].

We evaluated the association between the presence of symptoms and socio-demographic and clinical characteristics among those with a PCR-confirmed SARS-CoV-2 infection. However, we decided not to report the results because the number of observations for analysis was considerably reduced, resulting in unstable estimates.

Our study has several limitations. First, only 79.1% of the target population participated, which may have led to an overestimation or underestimation of the prevalence and incidence. Although no major differences were found between those who participated and those who did not regarding age, sex, or occupation, we did not have information about the history of infection in the non-participants; therefore, we cannot rule out selection bias.

Second, there could have been misclassification bias of SARS-CoV-2 infection considering the sensitivity and specificity of the HA test. To reduce false positives, we also performed other antibody tests in the case of a positive HA. Due to economic limitations in the availability of serologic diagnostic testing kits, we did not confirm the negative HA results with other serologic tests, which might have led to the underestimation of seroprevalence and cumulative incidence if the test used had imperfect sensitivity. We expect that the potential misclassification of the outcome be nondifferential. To examine the association of interest, we carried out

an analysis that was stringent regarding the classification of a case of infection using only confirmed positive HA results. To determine if the definition of the case affected the results, we conducted a sensitivity analysis with a flexible definition that used all positive HA results as cases, and we found consistent results.

Third, only 63.7% of the study population returned for the follow-up blood draw for the seroconversion study. Those who returned might have had a reduced risk of SARS-CoV-2 infection due to being older and less frequently involved with direct patient care, which could have led to an underestimation of the seroconversion frequency.

Lastly, although 97.4% of the HCWs reported using complete PPE, a thorough evaluation of PPE adherence was lacking, and we could not evaluate its association with the risk of infection. In addition, in our analytical model, we adjusted for the known or hypothesized risk factors that could be measured at the time of the study; however, some uncontrolled confounding may persist and explain part of the associations observed.

This study examined the dynamics of the SARS-CoV-2 infection in the first year of the pandemic among the HCWs of a tertiary referral hospital in Bogotá and found a high risk of infection, which reflects the situation in the community. Socio-economic vulnerability due to poverty and inequality in the social impact of the pandemic are factors linked to the exponential growth of COVID-19 in our country, especially in the pre-vaccination era [25, 45].

Our findings highlight the need to intensify efforts in prevention, education on the use of PPE, and detection of SARS-CoV-2 in HCWs, especially the front-line ones working in COVID-19 areas, emergency rooms, ICUs, or general wards; nurses and nurse assistants; those who are obese or overweight; and the young ones. This last group, despite having a lower risk of unfavorable outcomes play a key role in transmission and can be a source of infection for older HCWs and patients at high risk of SARS-CoV-2 complications.

This study is important because it generates knowledge about the burden of SARS-CoV-2 infection and risk factors for it among HCWs. Such knowledge contributes to the preservation of the wellbeing of healthcare personnel, which is essential for functioning healthcare systems that arecrucial for reducing the mortality and morbidity from the COVID-19 pandemic.

## Supporting information

**S1 Methods. Supplementary laboratory methods.**
(DOCX)

**S1 Dataset. Excel file with dataset used for the analysis and variables dictionary.**
(XLSX)

**S1 Fig. Testing of WHO reference serum with the Hemaglutination Assay (HA).** Titration of WHO reference serums 130, 120, 122, 124, 128, and negative serum samples with the HA. Starting with a 1/40 serum dilution (first column) serial dilutions were prepared of each serum up to a 1/20480 dilution. PBS was included in the last column. Blue circles indicate the last dilution with no tear indicating the titer of the serum.
(TIF)

**S2 Fig. Algorithm for definition of seropositive cases.** Individuals with a positive result of the HA and the indicated concomitant positive assays/conditions were considered seropositive (blue and light green colors). Individuals with a positive HA, but lacking supportive evidence for seropositivity (purple colors) were considered false positives and classified as seronegative.
(TIF)

**S3 Fig. Algorithm for classification of acute SARS-CoV-2 infections.** Among PCR positive cases, symptomatic individuals were considered acute infections. In asymptomatic HCWs, a person was considered acutely infected when they had a positive RT-PCR and any of the following conditions: negative HA, or positive HA and positive IgM by ELFA, or positive HA with negative IgM by ELFA and negative IgG by CLIA.
(TIF)

**S1 Table. Comparison of healthcare workers (HCWs) at Hospital Universitario San Ignacio who participated and did not participate in the study.** November 2020. [1]Column-based percentages.
(DOCX)

**S2 Table. Comparison of healthcare workers at Hospital Universitario San Ignacio who returned and did not return for the follow-up in the prospective cohort for studying seroconversion (December 15, 2020, to February 26, 2021).** [1]Column-based percentages. [2]ICU = Intensive Care Unit. [3]HCW who has worked in the COVID area sometime since March 2020. [4]HCW who was less than 6 feet away from an infected person (laboratory-confirmed or a clinical diagnosis) for a cumulative total of 15 minutes without personal protection elements sometime since March 2020. [5]Shared transportation was defined as the use of any public or collective transport. [6]History of smoking in the last year. [7]History of influenza vaccination in the last year. [8]Self-reported pre-existing medical condition.
(DOCX)

## Acknowledgments

To the HUSI HCWs who participated in the study.

To the clinical laboratory personnel who participated in the collection and processing of the samples required for the study.

We are grateful to Dr Tiong Kit Tan and Prof Alain Townsend for technical discussion and for supplying the HAT reagents for this study, and to the donors of the Townsend-Jeantet Prize Charitable Trust Charity No 1011770 for support. We would like to thank Dr. Scott Boyd from Stanford University for the gift of the RBD used as antigen in the ELISA.

To the Doctoral (PhD) Program in Clinical Epidemiology of the Faculty of Medicine, Pontificia Universidad Javeriana, Bogotá, Colombia, to which Sandra Liliana Valderrama-Beltrán is enrolled as a PhD candidate.

## Author Contributions

**Conceptualization:** Sandra Liliana Valderrama-Beltrán, Juliana Cuervo-Rojas, Beatriz Ariza, Claudia Cardozo, Juana Ángel, Magda Cepeda, Julio Cesar Castellanos, Carlos Gómez-Restrepo, Manuel Antonio Franco.

**Data curation:** Sandra Liliana Valderrama-Beltrán, Juliana Cuervo-Rojas, Samuel Martinez-Vernaza, María Juliana Soto, Julieth Arcila, Diana Salgado, Martín Rondón, Manuel Antonio Franco.

**Formal analysis:** Sandra Liliana Valderrama-Beltrán, Juliana Cuervo-Rojas, Juana Ángel, Samuel Martinez-Vernaza, María Juliana Soto, Magda Cepeda, Carlos Gómez-Restrepo, Manuel Antonio Franco.

**Funding acquisition:** Sandra Liliana Valderrama-Beltrán, Claudia Cardozo, Julio Cesar Castellanos, Carlos Gómez-Restrepo, Manuel Antonio Franco.

**Investigation:** Sandra Liliana Valderrama-Beltrán, Juliana Cuervo-Rojas, Juana Ángel, Julieth Arcila, Magda Cepeda, Carlos Gómez-Restrepo, Manuel Antonio Franco.

**Methodology:** Sandra Liliana Valderrama-Beltrán, Juliana Cuervo-Rojas, Martín Rondón, Magda Cepeda, Carlos Gómez-Restrepo, Manuel Antonio Franco.

**Project administration:** Sandra Liliana Valderrama-Beltrán, Beatriz Ariza, Samuel Martinez-Vernaza, Julieth Arcila, Diana Salgado.

**Resources:** Sandra Liliana Valderrama-Beltrán, Beatriz Ariza, Claudia Cardozo, Julieth Arcila, Diana Salgado, Carlos Gómez-Restrepo, Manuel Antonio Franco.

**Software:** Sandra Liliana Valderrama-Beltrán, Martín Rondón.

**Supervision:** Sandra Liliana Valderrama-Beltrán, Julio Cesar Castellanos, Manuel Antonio Franco.

**Validation:** Sandra Liliana Valderrama-Beltrán, Manuel Antonio Franco.

**Visualization:** Sandra Liliana Valderrama-Beltrán.

**Writing – original draft:** Sandra Liliana Valderrama-Beltrán, Juliana Cuervo-Rojas, Juana Ángel, María Juliana Soto, Manuel Antonio Franco.

**Writing – review & editing:** Sandra Liliana Valderrama-Beltrán, Juliana Cuervo-Rojas, Beatriz Ariza, Claudia Cardozo, Juana Ángel, María Juliana Soto, Diana Salgado, Martín Rondón, Julio Cesar Castellanos, Manuel Antonio Franco.

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
