## [Decision Letter · Decision Letter 0]

31 May 2022

PONE-D-22-11889Cumulative incidence, prevalence, seroconversion, and associated factors for SARS-CoV-2 infection among healthcare workers of a University Hospital in Bogota, ColombiaPLOS ONE

Dear Dr. Valderrama,

Thank you for submitting your manuscript to PLOS ONE. After careful consideration, we feel that it has merit but does not fully meet PLOS ONE’s publication criteria as it currently stands. Therefore, we invite you to submit a revised version of the manuscript that addresses the points raised during the review process.

We look forward to receiving your revised manuscript.

Kind regards,

Amitava Mukherjee, ME, Ph.D.

Academic Editor

PLOS ONE

Journal Requirements:

"This study was funded by Pontificia Universidad Javeriana, Hospital Universitario San Ignacio, and Fundación Bolívar Davivienda.

We are grateful to Dr Tiong Kit Tan and Prof Alain Townsend for technical discussion and for supplying the HAT reagents for this study, and to the donors of the Townsend-Jeantet Prize Charitable Trust Charity No 1011770 for support."

"This study was funded by Pontificia Universidad Javeriana, Hospital Universitario San Ignacio, and Fundación Bolívar Davivienda."

"This study was funded by Pontificia Universidad Javeriana, Hospital Universitario San Ignacio, and Fundación Bolívar Davivienda.

We are grateful to Dr Tiong Kit Tan and Prof Alain Townsend for technical discussion and for supplying the HAT reagents for this study, and to the donors of the Townsend-Jeantet Prize Charitable Trust Charity No 1011770 for support."

"None of the authors declare any conflict of interest for the conduction of this study."

Reviewers' comments:

Reviewer's Responses to Questions

**Comments to the Author**

1. Is the manuscript technically sound, and do the data support the conclusions?

Reviewer #1: Partly

2. Has the statistical analysis been performed appropriately and rigorously? 

Reviewer #1: Yes

3. Have the authors made all data underlying the findings in their manuscript fully available?

Reviewer #1: Yes

4. Is the manuscript presented in an intelligible fashion and written in standard English?

Reviewer #1: Yes

5. Review Comments to the Author

Reviewer #1: Authors have presented a manuscript entitled ´ Cumulative incidence, prevalence, seroconversion, and associated factors for SARS-CoV-2 infection among healthcare workers of a University Hospital in Bogota, Colombia ´ to be considered for publication in the journal Plos One.

The article has a very timely topic regarding the incidence and risk factors associated to SARS-CoV-2 infection among healthcare workers. However, the manuscript would benefit from a clearer design and dissemination of the results and conclusions. However, I have the following concerns about the manuscript, which should be addressed before publishing this work.

My main concerns are:

1. The authors present a legit conclusion, but the manuscript would benefit from a clear hypothesis. Particularly the authors are advised to address, what is the prevalence, incidence and likely epidemiological role of asymptomatic – either pre-symptomatic or asymptomatic – SARS-CoV-2 infections, particularly if nosocomially acquired, among the healthcare personnel in the University Hospital in Bogota. These results could be further compared to healthcare workers eg. in Sweden where very limited preventative measures were implemented. See Pimenoff et al. 2021 PLoS One. doi: 10.1371/journal.pone.0260453.

2. It is surprising to see weight associate to the risk of acquiring SARS-CoV-2 infection. However, this is likely a bias in the data as overweight is one of the risk factors for severe COVID-19 and thus overweight HCWs may be more willing to participate in SARS-CoV-2 infection screening than non-risk category HCWs. A systematic sensitivity analysis should be performed to identify if the dataset presented in this study is biased by selection of any severe COVID-19 disease related risk factors.

3. A figure of the prevalence data as a function of time would be a good way to visualize the data and related it to particular waves of the panemia and the dominant variant.

4. Manuscript text could have a bit more flow, revision of the text is advised.

6. PLOS authors have the option to publish the peer review history of their article (what does this mean?). If published, this will include your full peer review and any attached files.

Reviewer #1: No

---

## [Author Response · Author response to Decision Letter 0]

15 Jul 2022

July 15, 2022

Reviewer and Academic Editor

PLOS ONE

Dear reviewer and academic editor 

We want to thank the reviewer for his insightful and helpful comments. A detailed response to each and every comment is provided below in bold. In addition, please find a reply to each and every one of the “Journal requirements” also in bold.

Journal Requirements:

1. Please ensure that your manuscript meets PLOS ONE’s style requirements, including those for file naming. 

Answer: The manuscript and files have been adjusted following the style requirements. 

2. Please state what role the funders took in the study. 

Answer: The following statement is added in Financial Disclosure: The funders had no role in study design, data collection and analysis, decision to publish, or preparation of the manuscript. We added the financial disclosure statement in the cover letter as advised.

"This study was funded by Pontificia Universidad Javeriana, Hospital Universitario San Ignacio, and Fundación Bolívar Davivienda."

"This study was funded by Pontificia Universidad Javeriana, Hospital Universitario San Ignacio, and Fundación Bolívar Davivienda.

We are grateful to Dr Tiong Kit Tan and Prof Alain Townsend for technical discussion and for supplying the HAT reagents for this study, and to the donors of the Townsend-Jeantet Prize Charitable Trust Charity No 1011770 for support."

Answer: We added the following statement of financial disclosure in the cover letter: “This study was funded by Pontificia Universidad Javeriana, Hospital Universitario San Ignacio, and Fundación Bolívar Davivienda. The funders had no role in study design, data collection and analysis, decision to publish, or preparation of the manuscript.”

4. Please complete your Competing Interests on the online submission form to state any Competing Interests. 

Answer: The following statement was added in the Competing Interests section: The authors have declared that no competing interests exist. (Page 36 Line 544).

5. In your Data Availability statement, you have not specified where the minimal data set underlying the results described in your manuscript can be found. PLOS defines a study's minimal data set as the underlying data used to reach the conclusions drawn in the manuscript and any additional data required to replicate the reported study findings in their entirety. All PLOS journals require that the minimal data set be made fully available. 

Answer: We uploaded to the editorial manager as supporting information an Excel file with the minimal dataset of our study and the data dictionary. The data are free of personal identifiers. We cited as S2 minimal dataset in the supporting information section of the manuscript. We also provided this information in the cover letter. 

Reply to the Reviewer´s Comments to the Author:

1. Is the manuscript technically sound, and do the data support the conclusions?

Reviewer #1: Partly

2. Has the statistical analysis been performed appropriately and rigorously?

Reviewer #1: Yes

3. Have the authors made all data underlying the findings in their manuscript fully available?

Reviewer #1: Yes

4. Is the manuscript presented in an intelligible fashion and written in standard English?

Reviewer #1: Yes

My main concerns are:

1. The authors present a legit conclusion, but the manuscript would benefit from a clear hypothesis. Particularly the authors are advised to address, what is the prevalence, incidence and likely epidemiological role of asymptomatic – either pre-symptomatic or asymptomatic – SARS-CoV-2 infections, particularly if nosocomially acquired, among the healthcare personnel in the University Hospital in Bogota. These results could be further compared to healthcare workers eg. in Sweden where very limited preventative measures were implemented. See Pimenoff et al. 2021 PLoS One. doi:10.1371/journal.pone.0260453.

Answer: Our objectives were to determine SARS-CoV-2 infection cumulative incidence, prevalence, seroconversion, and associated factors among healthcare workers (HCWs) of a University Hospital in Bogota, Colombia and we constructed a model in which we evaluated different sociodemographic, clinical, and occupational characteristics and their association with the presence of SARS-CoV-2 infection. 

Furthermore, in a random subsample of HWCs in which RT-PCR was carried out to assess acute infection, we found a low prevalence (2.8%). Asymptomatic infections accounted for half of them (55%), highlighting the importance of screening as a containment measure in healthcare settings. 

In the discussion (Pages 30-31 Line 408 to 412) we now reference for comparison the study suggested by the reviewer, elaborating on the prevalence, incidence, and likely epidemiological role of asymptomatic – either pre-symptomatic or asymptomatic – SARS-CoV-2 infections, among healthcare personnel. 

However, we did not design the study to evaluate the role of asymptomatic and presymptomatic infections in the occurrence of new infections in our cohort as was done in 1. Pimenoff VN, Elfström M, Lundgren KC, Klevebro S, Melen E, Dillner J. Potential SARS-CoV-2 infectiousness among asymptomatic healthcare workers. PLoS One. 2021;16(12 December):1–7, 2. For this reason, we prefer not to formulate a hypothesis similar to what is presented in the cited paper. 

2. It is surprising to see weight associate to the risk of acquiring SARS-CoV-2 infection. However, this is likely a bias in the data as overweight is one of the risk factors for severe COVID-19 and thus overweight HCWs may be more willing to participate in SARS-CoV-2 infection screening than non-risk category HCWs. A systematic sensitivity analysis should be performed to identify if the dataset presented in this study is biased by selection of any severe COVID-19 disease related risk factors.

Answer: Since we did not include all of the target population in the study, we cannot exclude a selection bias related to being overweight or having any other associated disease or factor. However, the differential participation of individuals at higher risk of infection or severe disease (since the characteristic should also be a risk factor for infection to generate bias) could generate an overestimation of the prevalence and incidence of infection (as stated in the limitations of the study), but not necessarily an overestimation of the association between obesity and the risk of infection, as long as the contrast between the individuals in categories defined by weight be valid, that is, unconfounded. This is achieved if the categories have the same distribution of all other risk factors for infection. In our analytical model, we adjusted for the known or hypothesized risk factors that could be measured at the time of the study; however, some uncontrolled confounding may persist and explain part of the associations observed. To explain this, we included the previous sentence in the limitations of the study (Page 27 Line 423 to 426).

In addition, our conclusion is strengthened by the fact that we found a gradient in the risk of SARS-CoV-2 infection according to weight. Finally, a recent publication from a South African study found an independent association between obesity and overweight and acquiring SARS-CoV-2 infection (Reference 40 of the new version of the paper). To support this, we included the following sentence (Page 32-33 Line 457 to 460) “A study in South Africa found that overweight and obesity in HCWs were associated with SARS-CoV-2 infection in the multivariate analysis [40]”. (Stead D, Adeniyi OV, Singata-Madliki M, Abrahams S, Batting J, Jelliman E, et Al. Cumulative incidence of SARS-CoV-2 and associated risk factors among healthcare workers: a cross-sectional study in the Eastern Cape, South Africa. BMJ Open. 2022 Mar 18;12(3):e058761. doi: 10.1136/bmjopen-2021-058761.)

To address the possible overestimation of SARS-CoV-2 infection prevalence and incidence due to potential differential participation of obese/overweight HCWs in our study, we obtained from the Human Resources Office of HUSI the global information on comorbidities of its HCWs, which is estimated based on a convenience sample different to the one we used (see table below). The prevalence of overweight was 34.7%, and obesity was 11.3% in 2020. In our study, we found frequencies of overweight (n= 826; 32.89%) and obese (n= 185; 7.4%) HCWs lower than the estimated for the overall population of the hospital. Thus, it is unlikely that we have overestimated the SARS-CoV-2 prevalence and incidence due to this differential participation. 

Finally, the prevalences of other clinically relevant factors in the HUSI HCWs were: hypothyroidism 4.0%, arterial hypertension 2.3% and smoking 9.8%. In contrast, in our study the corresponding prevalences were 4.4%, 5.1% and 12.6%. Despite a slight overrepresentation of HCWs with arterial hypertension and smoking in our study, neither of them was significantly associated to an increased risk of acquiring SARS-Cov-2 infection. 

Severity Risk Factor Global estimated HUSI (%) Our Study (%)

Overweight 34.7 32.9

Obesity 11.3 7.4

Hypothyroidism 4.0 4.4

Arterial Hypertension 2.3 5.1

Smoking 9.8 12.6

3. A figure of the prevalence data as a function of time would be a good way to visualize the data and related it to waves of the pandemic and the dominant variant.

Answer: The figure proposed by the reviewer is cited in the manuscript (Page18 Line 288-291) as well as the figure caption (Page 23 Line 309-323). This figure will be uploaded as a separate file meeting the requirements. 

Fig 1. Seroprevalence and cumulative incidence of SARS-CoV-2 infection in Hospital Universitario San Ignacio (HUSI) healthcare workers (HCWs), and Bogota's COVID-19 epidemic curve. A. The SARS-CoV-2 seroprevalence in HUSI HCWs was 21.5% between November 17, 2020 and February 12, 2021 (n = 2,597) and 24.8% (n = 1,654) between December 15, 2020 and February 26, 2021. B. The SARS-CoV-2 epidemic curve of Bogota between March 2020 and February 2021 shows two epidemic waves. The first one began in June 2020 and ended approximately in October 2020, and the second one began in November 2020 and ended in February 2021. In this last epidemic wave, Gamma (P.1) and Mu (B.1.621) variants were introduced in the city. C. The SARS-CoV-2 cumulative incidence in HUSI HCWs was 35.7% (927/2,597) between March 6, 2020 and February 12, 2021. *The numbers of SARS-CoV-2 infection cases were taken from: https://saludata.saludcapital.gov.co/osb/index.php/datos-de-salud/enfermedades-trasmisibles/covid19/

4. Manuscript text could have a bit more flow, revision of the text is advised.

Answer: The suggested revision was carried out by an English editing service. We uploaded the revision certificate provided by this service.

---

## [Decision Letter · Decision Letter 1]

30 Aug 2022

Cumulative incidence, prevalence, seroconversion, and associated factors for SARS-CoV-2 infection among healthcare workers of a University Hospital in Bogotá, Colombia

PONE-D-22-11889R1

Dear Dr. Valderrama,

We’re pleased to inform you that your manuscript has been judged scientifically suitable for publication and will be formally accepted for publication once it meets all outstanding technical requirements.

Kind regards,

Amitava Mukherjee, ME, Ph.D.

Academic Editor

PLOS ONE

Additional Editor Comments (optional):

Reviewers' comments:

Reviewer's Responses to Questions

**Comments to the Author**

1. If the authors have adequately addressed your comments raised in a previous round of review and you feel that this manuscript is now acceptable for publication, you may indicate that here to bypass the “Comments to the Author” section, enter your conflict of interest statement in the “Confidential to Editor” section, and submit your "Accept" recommendation.

Reviewer #1: All comments have been addressed

2. Is the manuscript technically sound, and do the data support the conclusions?

Reviewer #1: Yes

3. Has the statistical analysis been performed appropriately and rigorously? 

Reviewer #1: Yes

4. Have the authors made all data underlying the findings in their manuscript fully available?

Reviewer #1: Yes

5. Is the manuscript presented in an intelligible fashion and written in standard English?

Reviewer #1: Yes

6. Review Comments to the Author

Reviewer #1: (No Response)

7. PLOS authors have the option to publish the peer review history of their article (what does this mean?). If published, this will include your full peer review and any attached files.

Reviewer #1: No

---

## [Editor Report · Acceptance letter]

9 Sep 2022

PONE-D-22-11889R1 

Cumulative incidence, prevalence, seroconversion, and associated factors for SARS-CoV-2 infection among healthcare workers of a University Hospital in Bogotá, Colombia 

Dear Dr. Valderrama-Beltrán:

I'm pleased to inform you that your manuscript has been deemed suitable for publication in PLOS ONE. Congratulations! Your manuscript is now with our production department. 

Kind regards, 

on behalf of

Professor Dr. Amitava Mukherjee 

Academic Editor

PLOS ONE